# Improved *Medicago sativa* Nodulation under Stress Assisted by *Variovorax* sp. Endophytes

**DOI:** 10.3390/plants11081091

**Published:** 2022-04-17

**Authors:** Noris J. Flores-Duarte, Julia Pérez-Pérez, Salvadora Navarro-Torre, Enrique Mateos-Naranjo, Susana Redondo-Gómez, Eloísa Pajuelo, Ignacio D. Rodríguez-Llorente

**Affiliations:** 1Departamento de Microbiología y Parasitología, Facultad de Farmacia, Universidad de Sevilla, 41012 Sevilla, Spain; nflores@us.es (N.J.F.-D.); jperper@ibmcp.upv.es (J.P.-P.); epajuelo@us.es (E.P.); 2Departamento de Biología Vegetal y Ecología, Facultad de Biología, Universidad de Sevilla, 41012 Sevilla, Spain; emana@us.es (E.M.-N.); susana@us.es (S.R.-G.)

**Keywords:** ACC deaminase, legume nodulation, plant-growth-promoting endophytes, degraded soils

## Abstract

Legumes are the recommended crops to fight against soil degradation and loss of fertility because of their known positive impacts on soils. Our interest is focused on the identification of plant-growth-promoting endophytes inhabiting nodules able to enhance legume growth in poor and/or degraded soils. The ability of *Variovorax paradoxus* S110^T^ and *Variovorax gossypii* JM-310^T^ to promote alfalfa growth in nutrient-poor and metal-contaminated estuarine soils was studied. Both strains behaved as nodule endophytes and improved in vitro seed germination and plant growth, as well as nodulation in co-inoculation with *Ensifer medicae* MA11. *Variovorax* ameliorated the physiological status of the plant, increased nodulation, chlorophyll and nitrogen content, and the response to stress and metal accumulation in the roots of alfalfa growing in degraded soils with moderate to high levels of contamination. The presence of plant-growth-promoting traits in *Variovorax*, particularly ACC deaminase activity, could be under the observed in planta effects. Although the couple *V. gossypii*-MA11 reported a great benefit to plant growth and nodulation, the best result was observed in plants inoculated with the combination of the three bacteria. These results suggest that *Variovorax* strains could be used as biofertilizers to improve the adaptation of legumes to degraded soils in soil-recovery programs.

## 1. Introduction

Climate change, industrial globalization, and agricultural activities have resulted in a worrying increase of poor or degraded soils, with low nutrient contents and contaminants limiting plant growth. In this scenario, one of the global challenges of this century is the sustainable use and recovery of soils to feed an increasing population [1]. Soil sustainability implies the development of green practices aimed at maintaining an ecological balance and enhancing soil functions and biodiversity. Recommended technologies of cropping systems should include legumes, whose positive impacts are well-known: biological nitrogen fixation that improves soil health and fertility, erosion control (as a cover crop), and a high nutritional value (grain legumes contain from 20–40% protein) [2].

Legumes are able to establish a well-studied symbiotic interaction with several genera of soil nitrogen-fixing bacteria commonly called rhizobia, which results in the formation of the nodule, a new organ where atmospheric nitrogen fixation takes place [3]. A large proportion of the nitrogen requirements of crops is provided through this symbiosis [4]. For decades, the scientific community has undertaken numerous efforts to identify the most suitable rhizobia–legume couples to obtain the best performance at the level of grain production according to the soil conditions. Later on, attempts to improve legume growth and nodulation using plant-growth-promoting bacteria (PGPB) in co-inoculation with suitable rhizobia have been made [5]. Recently, Martinez-Hidalgo and Hirsch [6] proposed the existence of a nodule microbiome, composed of a variable number of non-rhizobial endophytes (NRE) [7] or nodule-associated bacteria (NAB) that do not induce nodule formation but might be able to colonize the nodules [8]. Bacteria belonging to a variety of genera, including among others *Acinetobacter*, *Agrobacterium*, *Bacillus*, *Enterobacter*, *Micromonospora*, *Mycobacterium*, *Paenibacillus*, *Pseudomonas*, and *Variovorax*, have been isolated from nodules of legumes [6,9]. The fact that the rhizobia are not alone in the nodule raises new questions, particularly the role of NAB as PGPB in enhancing legume growth and nodulation.

Several strains from the *Variovorax* genus have been isolated from nodules of wild-type legumes growing in arid regions around the world. *Variovorax paradoxus* XB3 was isolated from nodules of *Crotalaria incana* growing in Ethiopia [10] and *V. paradoxus* DSM66^T^ and *Variovorax* sp. MN36.2^T^ were isolated from nodules of *Acacia* in Australia [11]. In the same way, *Variovorax* sp. CT7.15 was isolated from *Calicotome villosa* nodules collected in arid zones of Tunisia [12]. This strain showed several plant-growth-promoting (PGP) properties, such as phosphate solubilization, 1-aminocyclopropane-1-carboxylic acid (ACC) deaminase activity, and enhanced *Calicotome* growth and nodulation in co-inoculation with wild-type rhizobia in arid Tunisian soils. *Variovorax* sp. CT7.15 was taxonomically related with *V. paradoxus*- and *V. gossypii*-type strains.

This work is aimed at assessing the potential of two endophytic *Variovorax*-type strains, *V. paradoxus* S110^T^ and *V. gossypii* JM-310^T^, to enhance legume growth and nodulation rates in poor, degraded, and/or metal-contaminated soils to facilitate legume adaptation and soil regeneration.

## 2. Results

### 2.1. Characterization of Strains

The tolerance of strains toward As, Cd, Cu, and Zn, recorded as the maximum tolerable concentration, was evaluated (Table 1). Variovorax strains tolerated higher concentrations, except for As, than E. medicae MA11. The three strains showed high levels of tolerance toward the assayed metal/loids.

Concerning PGP properties, MA11 was able to solubilize phosphate, produced siderophores and IAA, formed biofilms, and had ACC deaminase activity (Table 2). *V. paradoxus* showed all the properties assayed, except for phosphate solubilization. *V. gossypii* did not solubilize phosphate nor form biofilms, being positive for the rest of the properties assayed (Table 2). The three strains were able to grow in a nitrogen-free medium and MA11 reduced acetylene.

The presence of enzyme activities in the strains was also examined (Table 2). The three strains showed amylase activity. *V. gossypii* was also positive for lipase, pectinase, and protease. Chitinase and cellulase were also assayed, but the three strains lacked these activities. 

PGP properties and enzyme activities in the presence of metal were also examined. The strains not only maintained the PGP properties in the presence of metals but some properties even became positive or certain levels increased (Table 2). In that way, both *Variovorax* strains were able to solubilize phosphate and *V. gossypii* formed biofilms in the presence of metals. Interestingly, all the strains significantly increased their levels of ACC deaminase activity. Nevertheless, the presence of metal induced the loss of most enzyme activities. *V. paradoxus* lost the amylase activity and MA11 kept it, while *V. gossypii* conserved only the protease activity.

### 2.2. Inoculation Enhanced M. sativa Seeds Germination

*M. sativa* seeds were germinated in the presence and absence of a mix of metals and As, and inoculated with single strains, co-inoculated with MA11 and *Variovorax* strains, or the three strains together (Figure 1). Seed inoculation enhanced germination both in the absence (Figure 1a) and presence of metals (Figure 1b). The highest increase was found in seeds inoculated with the combination of the three strains (CSV), showing 40.6% and 37% more germinated seeds than the non-inoculated control in the presence and absence of metals, respectively. The lowest increase was recorded when seeds were inoculated with MA11. The increase in the germination rate in the presence and absence of metals followed the pattern: CSV > Vg + MA11 > Vp + MA11 > Vg > Vp > MA11 > C–. 

### 2.3. Variovorax Increased M. sativa Growth and Nodulation In Vitro

To assess the ability of the *Variovorax* strains to increase *M. sativa* growth and nodulation, seedlings were inoculated with MA11, MA11 + Vg, MA11 + Vp, or the consortium combining the three strains and placed on plates with or without As. Non-inoculated seedlings were used as control (Figure 2). As expected, *M. sativa* plants reached lower values of weight in the presence than in the absence of arsenic, independent of the inoculation condition. Plants inoculated with MA11 showed higher dry weight than the control plants, although the differences were significant only in the shoots (Figure 2a,b). Co-inoculation with the *Variovorax* strains and MA11 induced higher plant dry weight both in roots and shoots than a single inoculation with MA11, with significant differences in the roots. Plants inoculated with the consortium showed significantly higher weight values than plants inoculated with MA11 or co-inoculated with rhizobia and the *Variovorax* strains, both in the presence and absence of metals (Figure 2a,b). Concerning nodulation rates, co-inoculation with any of the *Variovorax* strains significantly increased the number of nodules compared with a single inoculation with MA11, both in the absence and presence of As (Figure 2c). While no differences between the *Variovorax* strains were observed in the absence of As, significant differences in nodule number appeared in the presence of the metalloid, the number of nodules being higher in plants co-inoculated with *V. gossypii* than those co-inoculated with *V. paradoxus*. Plants inoculated with the combination of the three strains recorded the highest number of nodules, 1.75-fold and 4-fold the number of nodules induced by MA11 alone in the absence (Figure 2c) and presence of As (Figure 2d), respectively.

### 2.4. Variovorax Behaves as Nodule Endophytes

Using confocal laser scanning microscopy (CLSM), mCherry-labeled *V. gossypii* and *V. paradoxus* cells were visualized from 0.5 mm slices of *M. sativa* roots and nodules. An example of the results obtained is presented in Figure 3. Although plant tissues exhibited green fluorescence, red fluorescent bacteria could be observed in both the root (Figure 3a) and nodule (Figure 3b) cells. Particularly, groups of bacteria were clearly appreciated in the nitrogen fixation zone of *M. sativa* nodules, marked with a red square in the figure (Figure 3b).

### 2.5. Variovorax Increased M. sativa Growth and Nodulation in Estuarine Soils with Nutrient Poverty

The effect of *M. sativa* inoculation with the *Variovorax* strains and/or MA11 in soils with a low content of organic matter and nutrients was evaluated in pots containing sterilized soil from the Odiel River estuary (OHM). Characteristics of the soil are presented in Table 3. 

A single inoculation with any of the strains increased plant biomass, both root and shoot biomasses, compared with the control plants (Figure 4a). Inoculation with the *Variovorax* strains reported higher values of root and shoot dry weight than inoculation with MA11. Particularly, plants inoculated with Vg showed higher biomass than plants inoculated with MA11, with significant differences. Co-inoculation with MA11 and Vg or Vp increased plant biomass compared to a single inoculation, the plants co-inoculated with MA11 and Vg showing higher dry-weight values than those co-inoculated with MA11 and Vp. Nevertheless, plants inoculated with the consortium of the three strains (CSV) showed the highest values of dry weight, with significant differences compared to the rest of the inoculation conditions (Figure 4a). In a similar way, plants co-inoculated with MA11 and the *Variovorax* strains reported higher root and shoot length than plants inoculated with a single bacterium and the highest values were measured in plants inoculated with the consortium (Figure 4b). Plants inoculated with the combination of the three strains also showed the highest number of leaves with the largest diameter, with significant differences compared to the other inoculation conditions (Appendix A). Regarding nodule numbers, the largest amount was again counted in plants inoculated with the consortium of the three strains (Figure 4c), which showed about 80% more nodules than plants inoculated with MA11. Plants inoculated with the MA11–Vg couple showed more nodules than those inoculated with MA11–Vp and, in both cases, co-inoculation (Variovorax–MA11) increased the number of nodules compared to a single inoculation with MA11. The differences were statistically significant (Figure 4c). The nitrogen content in plant leaves and stems was also estimated (Figure 4d). A single inoculation with any of the bacteria increased N content compared with non-inoculated plants. Surprisingly, plants inoculated with *Variovorax* reported significantly higher N values than those inoculated with rhizobia, while co-inoculation with *Variovorax* and MA11 did not ameliorate the results obtained with *Variovorax* as a single inoculant. Finally, the highest N content was found in plants inoculated with the consortium containing the three strains, with significant differences.

Concerning chlorophyll content, the inoculated plants had significantly higher values than the control plants. The total chlorophyll content in the inoculated plants could be described with the following pattern, with significant differences (Figure 5a): CSV > Vg + MA11 > Vp + MA11 = Vg > Vp > MA11 > C–.

The response of the plants to stress was analyzed by recording the activity of several plant enzymes related to the activation of the antioxidant mechanisms in the plant (Figure 6). In general, inoculation increased enzymatic activities, particularly with significant differences in the roots. The highest levels of ascorbate peroxidase activity could be observed in the roots and shoots of plants inoculated with the consortium (Figure 6a). The levels of catalase activity were similar in the roots of all the inoculated plants, independent of the inoculation conditions (Figure 6b), while this activity was higher in the shoots of plants inoculated with the consortium or the combination of MA11 with *V. paradoxus* than in the shoots of plants inoculated with other bacteria (Figure 6b). Guaiacol peroxidase and superoxide dismutase activities showed a similar behavior, with the highest activities recorded in the roots and shoots of the plants inoculated with the consortium (Figure 6c,d).

### 2.6. Variovorax Increased M. sativa Growth and Nodulation in Estuarine Soils Contaminated with Metals

*M. sativa* seedlings were placed in pots containing a mix of estuarine soils contaminated with metals and inoculated with MA11 or the bacterial consortium containing *Variovorax* strains and MA11 (CSV). For soil preparation, three parts of OHM soil was mixed with one part of OLM soil, the characteristics of which are summarized in Table 3. Inoculated plants showed higher biomass and root and shoot length than non-inoculated plants (Figure 7a,b). Significant differences between the plants inoculated with MA11 and those inoculated with the consortium were found in all the parameters recorded (Figure 7 and Appendix A). The shoots of plants inoculated with the *consortium* had 2.5-fold the biomass reached by the plants inoculated with MA11 (Figure 7a). Plants inoculated with the consortium also had more root biomass (Figure 7a), longer roots and shoots (Figure 7b), and more leaves with larger diameters (Appendix A) than the plants inoculated with MA11.

The bacterial consortium induced 3.5-fold the number of nodules elicited by MA11 (Figure 7c). About one-third of these nodules were small and non-well-developed. These differences were reflected in the shoots’ nitrogen content (Figure 7d). Plants inoculated with the consortium had 20% more nitrogen than the plants inoculated with MA11 as a single inoculant.

Regarding total chlorophyll content, inoculation with MA11 provoked an increase in chlorophyll content compared to that of the non-inoculated control plants and the highest content was again recorded in the plants inoculated with the consortium, with statistically significant differences (Figure 5b).

Finally, enzymatic activities related to plant-stress management increased in the inoculated plants (Figure 8). The highest levels of ascorbate peroxidase, catalase, and guaiacol peroxidase activities were recorded in the roots and shoots of plants inoculated with the consortium (Figure 8a–c). Concerning superoxide dismutase activity, no differences among the inoculation conditions were found in shoots, while the highest activity in roots was again recorded in the plants inoculated with the consortium (Figure 8d).

### 2.7. Variovorax Increased Metal Accumulation on M. sativa Roots

Metal accumulation in *M. sativa* tissues was measured in plants growing in OHM and the mix of both soils (Figure 9 and Table 4). The highest levels of metal/loids found in the shoots of *M. sativa* plants growing in soils poor in nutrients were 4.8 ppm of As, 0.15 ppm of Cd, 36.2 ppm of Cu, and 60 ppm of Zn, without significant differences among the inoculation conditions. Nevertheless, significant differences were observed in the amounts of metals that accumulated in the roots of *M. sativa* depending on the inoculation treatment (Figure 9). Inoculation increased As and metals accumulation and the highest values were always observed in the roots inoculated with the consortium (Figure 9a–d). In general, roots inoculated with *V. gossypii* or its combination with MA11 reported higher values of As, Cd, Cu, and Zn than roots inoculated with *V. paradoxus* or its combination with MA11 (Figure 9a–d).

Regarding plants growing in soils contaminated with metals, inoculation increased As and metal accumulation both in the roots and the shoots (Table 4). The highest levels of As, Cd, and Zn accumulation in shoots were found in the plants inoculated with MA11, while the plants inoculated with the consortium showed higher concentrations of Zn in shoots than the plants inoculated with MA11. Concerning As and metals accumulation in roots, the highest concentrations were found in the roots inoculated with the consortium, with significant differences (Table 4).

## 3. Discussion

If we want to feed the increasing world population, it is mandatory to control soil degradation and loss of fertility to ameliorate the quantity and quality of our crops. The sustainable use of soils and their recovery for agricultural purposes should involve legumes because of their known beneficial effects for soils [2]. Traditionally, the main tool to promote legume growth and adaptation has been the selection of the most adequate rhizobia as an inoculant for nodulation in every soil condition. We propose that this selection should be accompanied by the identification of nodule endophytes able to enhance plant nodulation and growth, resulting in better plant adaptation to stressing soil conditions. 

Based on this premise, we isolated and characterized the endophyte *Variovorax* sp. CT7.15 from nodules of *C. villosa*, a legume naturally growing in the arid soils of Tunisia [12]. This strain behaved as a nodule-enhancing bacteria (NEB) in arid soils in co-inoculation assays with different autochthonous rhizobia [11]. Since strain CT7.15 is taxonomically located between *V. paradoxus* and *V. gossypii*, these species could also behave as NEB. This work was aimed at evaluating the ability of these species to enhance legume growth and adaptation in stressing soil conditions, such as nutrient poverty and/or the presence of metals. *V. paradoxus* is a metabolically versatile species that has been isolated from a great variety of ecosystems all around the world, both as free-living bacteria or plant endophytes [13]. *V. gossypii* was isolated as an endophyte from the internal root tissues of *Gossypium hirsutum* [14]. 

Characterization of *V. paradoxus* S110^T^ and *V. gossypii* JM-310^T^ revealed their resistance to high concentrations of As, Cd, Cu, and Zn, and the presence of several PGP properties that were, in general, maintained or increased in the presence of these metal/loids. We chose from our collection *Ensifer medicae* MA11 as the rhizobia to induce *M. sativa* nodulation, a strain resistant to metals and able to induce nodulation in stressing conditions [15].

Our results showed that both *Variovorax* strains promoted *M. sativa* growth in vitro and in greenhouse conditions in soil, from germination to plant nodulation, as a single inoculant or in co-inoculation with the rhizobia. Both strains were localized as endophytes inside the roots and nodules of alfalfa plants inoculated with MA11.

*E. medicae* MA11 and both *Variovorax* strains enhanced seed germination in the presence and absence of metals. One bacterial property that could ameliorate germination is IAA production, a hormone that acts as a coordinator of plant development and triggers seed germination [16]. The effect of the auxin produced by bacteria on seed germination could be particularly relevant in the presence of metals [17]. The *V. paradoxus* S110^T^ (genome BCUT01) and *V. gossypii* JM-310^T^ (genome RXOE01) genomes have genes involved in IAA synthesis and both strains produced IAA in the presence and absence of metals. Since starch degradation is an important source of energy during seed germination for some plant species, bacterial amylase activity could also be involved in ameliorating this early stage [18]. Finally, *V. gosypii* had pectinase activity, which could also facilitate seed germination through pectin degradation during this process [19].

Concerning the positive effect of both *Variovorax* in plant growth and nodulation in vitro, it is important to notice that legume nodulation is inhibited by the ethylene produced by plants growing on plates, with limiting amounts of oxygen and some nutrients in long-term experiments, and with arsenic as an additional stressing factor [20,21]. It is also known that inhibitors of ethylene biosynthesis have a positive effect on nodulation [22,23]. At this point, the presence of ACC deaminase activity in the *Variovorax* and MA11 strains allows bacteria to modulate the ethylene concentration in the plants, since this enzyme breaks an ethylene precursor [24,25]. Thanks to this modulation, bacteria allow plant nodulation and growth under stressing conditions [26]. In addition, bacterial IAA could stimulate the absorption of root-elongation-facilitating nutrients [27]. Although both *Variovorax* strains showed ACC deaminase activity, the *acdS* gene coding this enzyme is only found in the *V. paradoxus* genome. 

Following this reasoning, the presence of ACC deaminase activity and IAA production would be particularly relevant in pot experiments with soil containing moderate to high levels of metals, nutrient poverty, and longer periods of plant growth and nodulation. These properties could, in addition, delay nodule senescence, improving nodule functionality since ethylene is also involved in nodule senescence [6,28,29] and IAA interferes in the rhizobia–legume dialogue interaction, delaying nodule senescence by interacting with the bacteroid inside the nodules [30]. In these soils, bacterial properties related with nutrient acquisition could be involved in ameliorating plant growth. Bacterial phosphate solubilization could provide assimilable phosphate for the plant, one of the major limiting nutrients in these soils [31]. Interestingly, this property was induced in *Variovorax* strains in the presence of metals. Siderophores produced by bacteria have a great affinity for iron, forming a complex that can be assimilated by plants [32,33]. Finally, although the *Variovorax* were able to grow in media without nitrogen and reduced small amounts of acetylene, both strains lack the nitrogenase genes (*nifKDH*), suggesting that they do not fix nitrogen. ACC deaminase is responsible for the cleavage of the plant ethylene precursor, ACC, into ammonia and α-ketobutyrate [34] and this ammonia produced by endophytes inside nodules and roots could explain the increase in nitrogen levels observed in the plants inoculated with *Variovorax*. The beneficial effect of *Variovorax* in plant nodule numbers could also explain the high levels of nitrogen measured in the plants inoculated with the consortium.

Stress conditions induce generation in the plant or the reactive oxygen species (ROS) and, as a consequence, cause oxidative damage, which affects plant growth [35]. To maintain homeostasis in high levels of ROS, plants can modulate the activity of antioxidative enzymes, such as ascorbate peroxidase, catalase, guaiacol peroxidase, and superoxide dismutase [35,36]. In this work, the increase of these enzymatic activities correlates with a better plant response to stress caused by nutrient deficiency and/or metals, activating the antioxidant mechanisms in the plant. Such an effect has been particularly observed in plants inoculated with combinations of MA11 and *Variovorax*. An increase in antioxidant enzymatic activities was also reported in *M. sativa* plants inoculated with rhizobacteria under salinity stress [37,38].

The effects of the As and Cu in *M. sativa* nodulation previously reported have been observed in this work, that is, the reduction in nodule number induced by As [15] and the presence of small non-developed nodules in the presence of Cu [39]. Inoculation of alfalfa with the combination of MA11 and *Variovorax* partially alleviated these effects, increasing the number of nodules found in plants growing in the presence of metals and As. Concerning metal accumulation, it has been largely described that legumes are able to accumulate metals in their roots, with low levels of metal translocation to the shoots [15,40]. Although we propose the use of legumes as pioneer plants to increase soil fertility and not for food or fodder, it is important to monitor metal accumulation in shoots to avoid metal transfer to the trophic chain. Toxicity limits in shoots for domestic animals were set at 30 ppm As, 10 ppm Cd, 40 ppm Cu, and 500 ppm Zn [41]. In this work, alfalfa plants accumulated in their shoots amounts of As, Cd, and Zn that were well below these limits. Nevertheless, Cu levels were close to 40 ppm in soils with moderate levels of metals and more than this value in soils with higher amounts of Cu. These results pointed to the need to determine metal accumulation in legume shoots growing in highly contaminated soils or with high metal availability prior to their use. In the other way, plants increased their capacity to accumulate metals in their roots after inoculation with *Variovorax*, particularly with *V. gossypii*, alone or combined with MA11. Nevertheless, the highest values of metals were recorded in roots inoculated with the consortium containing the three strains. Siderophores produced by *Variovorax* could promote a plant’s metal uptake, unless it is not clear if siderophores promote or reduce a plant’s metal uptake and there are examples of both behaviors [42]. These results suggest that a combined inoculation with rhizobia and *Variovorax* increased the metal phytostabilization potential of alfalfa plants, by increasing root development and root metal uptake.

In summary, positive effects on plant growth, nodulation, response to stress, and metal accumulation in the roots of *M. sativa* after inoculation followed the pattern: CSV > Vg + MA11 > Vp + MA11 > Vg > Vp > MA11 > C–. Although the combination of *V. gossypii* and MA11 reported a great benefit to plant growth and nodulation, an extra benefit could be observed when *V. paradoxus* was included in the inoculant, indicating that the use of a consortium can report better results than individual inoculation because of the combined effect of the bacteria. It is important to notice that *V. paradoxus* and *V. gossypii* (in that case only in the presence of metals) were able to form biofilms that could help plants to grow in degraded environments by concentrating bacteria in the roots and facilitating nutrient mineralization and absorption [43].

## 4. Materials and Methods

### 4.1. Bacterial Strains and Metal Tolerance

*Variovorax paradoxus* S110^T^ and *V. gossypii* JM-310^T^ were obtained from the Belgian Coordinated Collection of Microorganisms (BCCM). *Ensifer medicae* MA11 was isolated by our research group from metal-contaminated soils of southwest Spain [15].

The tolerance of these strains toward heavy metals and sodium arsenite was assayed on TSA (tryptic soy agar) plates for *Variovorax*, or TY plates [44] for *Ensifer*, supplemented with increasing concentrations of metal/loid using the following stock solutions: NaAsO_2_ 0.5 M, CuSO_4_ 1 M, CdCl_2_ 1 M, ZnSO_4_ 1 M. Metal tolerance was expressed as the maximum tolerable concentration (MTC), that is, the highest metal/loid concentration allowing bacterial growth.

### 4.2. Determination of PGP Properties 

As a preliminary test for nitrogen fixation, strains were plated on nitrogen-free broth (NFB) medium [45]. Strains able to grow in NFB were then tested for acetylene reduction as described in [46]. Phosphate solubilization was detected on NBRIP medium plates [47] when bacterial growth provoked the formation of transparent halos around the bacterial colonies after 72 h of incubation at 28 °C. Similarly, orange halos indicated siderophore production on CAS (chrome azurol S) [48] agar plates after 72 h of incubation at 28 °C. Indole-3-acetic acid (IAA) production was evaluated by a colorimetric technique from a liquid culture supplemented with L-tryptophan (0.1 mg/mL) and incubated at 28 °C. After incubation, the Salkowski reagent [49] was added and a pink color developed in the positive results, which were measured at OD_530_ using a spectrophotometer (Lambda 25; PerkinElmer, Walthmam, MA, USA). The results were compared with a calibration curve of pure IAA as a standard following the linear regression analysis. Measures were taken at 72 h of incubation. The presence of the ACC deaminase enzyme was determined using a modification of the Penrose and Glick method [50] detailed in [51]. Briefly, strains were plated in solid Dworkin and Foster (DF) mineral medium [52] with 3.0 mM ACC after enrichment in a solid DF salts minimal medium with (NH_4_)_2_SO_4_ as the source of the nitrogen. The ACC solution was thawed, then added to the sterile DF medium and spread on plates. ACC was allowed to fully dry before testing individual colonies. A DF medium with no N source was used as the control. The growth on the plates was checked daily during five days at 28 °C. The ACC deaminase activity was measured by monitoring the amount of α-ketobutyric acid generated from the cleavage of ACC [50]. The reaction was determined by comparing the absorbance at 540 nm of the sample to a standard curve of α-ketobutyrate. After determining the amount of protein, using the Bradford method [53], and α-ketobutyrate, the enzyme activity was calculated based on the μmoles of released α-ketobutyrate per mg of protein per hour. The capability of biofilm formation was measured by an adhesion capacity assay in wells of 96-wellmicrotitre plates exactly as described in [54].

Strains isolated and characterized in previous works were used as the positive and negative controls [51,54].

### 4.3. Screening for Bacterial Enzyme Activities

Enzyme activities were tested on plates that were incubated for 5–7 days at 28 °C. Strains were plated on starch agar (Scharlab, Barcelona, Spain) to detect amylase activity and revealed after incubation by flooding with lugol (iodine potassium iodine solution). To test cellulase activity, strains were plated onto solid M9 minimal medium supplemented with a yeast extract (0.2%) and carboxymethyl cellulose (1%). Plates were revealed after incubation by flooding with Congo red solution 1mg/mL for 15 min and destaining with 1 M sodium chloride for 15 min [55]. For pectinase activity, ammonium mineral agar (AMA) plates were used. Plates were revealed with 2% CTAB and positive bacteria showed a halo around [48]. Protease activity was assayed by growing the strains in casein agar [56]. Plates were incubated and observed for clear zones around the cultures. Strains were grown in Tween agar [56] for lipase activity. A precipitate around the strains could be observed as a positive result. Chitinase activity was tested using the minimal medium described in [51] supplemented with colloidal chitin (1.5%), with zones of chitin clearing around the colonies after incubation. 

Strains isolated and characterized in previous works were used as the positive and negative controls [51,54].

### 4.4. PGP Properties and Enzyme Activities in Presence of Metals

PGP properties and enzyme activities were also assayed in the presence of a mix of metal/loids containing 0.3 mM of As, Cu, and Zn and 0.05 mM of Cd. The appropriate amounts of NaAsO_2_ 0.5 M, CuSO_4_ 1 M, CdCl_2_ 1 M, and ZnSO_4_ 1 M stock solution were added aseptically to the corresponding medium. The production of siderophores in the presence of metals could not be recorded since it is not possible to prepare CAS media in their presence because of metal precipitation.

### 4.5. Labelled of Variovorax with Fluorescence and Microscopy

*Variovorax* strains were labeled with the fluorescent protein mCherry by bacterial conjugation, mixing the donor *Escherichia coli* DH5α containing plasmid *pMP7604* [57], the helper *E. coli* strain containing *pRK600* [58], and the recipient *Variovorax* strain in liquid TSB medium by slightly shaking at 28 °C for 16 h. Then, 100 µL aliquots were plated onto TSA plates containing rifampicin (100 µg/mL) as a selective antibiotic for *Variovorax* and tetracycline (10 µg/mL) for *pMP7604* plasmid. *M sativa* plants were co-inoculated with *E. medicae* MA11 and each of the labeled strains for nodule induction on squared plates as described below (20 seedlings per condition). After 28 days of incubation, the roots and nodules were excised and 0.5 mm handmade cuttings were observed [51]. Fluorescent bacteria in plant tissues were visualized using a laser scanning spectral confocal optical microscope (Zeiss LSM 7 DUO, Zeiss, Jena, Germany) with an objective Plan-Apochromat 20X/0.8 M27, filters of 572–727, and a laser of 561 nm (5.3%). Images were processed with ZEN2011 software (Zeiss, Jena, Germany).

### 4.6. Medicago sativa Seeds Germination and Growth on Plates

Seeds of alfalfa (*M. sativa* L. ecotype, Aragon) were surface-disinfected on 70% ethanol for 10 min. Next, they were placed in 5% sodium hypochlorite for 30 min with gentle shaking and washed six times with sterile distilled water and placed in water-agar plates (1%). After disinfection, seeds were submerged for 1 h in 5 mL of cultures comprising the desired bacteria (10^8^ cells/mL) [59]. The cultures were prepared by growing the strains in TSB or TY for 24 h at 28 °C, then centrifuged and resuspended in 0.9 g/L NaCl solution. Control seeds were submerged in a 5 mL 0.9 g/L NaCl solution. The seeds were then transferred to agar-water plates containing a mix of 7.5 μM of As, Cd, Cu, and Zn, prepared from the stock solutions described above. Plates without metals were used as the control. Inoculation conditions were control without inoculation (C–), *E. medicae* MA11 (MA11), *V. paradoxus* (Vp), *V. gossypii* (Vg), co-inoculated with *E. medicae* MA11 and *V. paradoxus* (MA11 + Vp) or *V. gossypii* (MA11 + Vg), or inoculated with the three strains (CSV). Five plates and 10 seeds per plate for each condition were used. Plates were incubated in the dark at 28 °C and germination was observed every 24 h.

For plant assays on square plates (12 × 12 cm), the 24 h pre-germinated seeds were transferred to plates containing a slope of BNM-agar medium [52], supplemented with 1 mM NH_4_NO_3_, containing 30 μM sodium arsenite or without arsenite. The seeds were inoculated with MA11 or co-inoculated in the conditions described above. Non-inoculated seeds were used as the control. Fifty seeds per condition (five plates, 10 seeds per plate) were used. The roots were preserved from light by covering the upper part of the plates with black paper [15]. Square plates were placed in an upright position and incubated in a growth chamber (AGP-700-HR ESP; Radiber, Barcelona, Spain) at 22 °C with an 8 h dark:16 h light cycle. Plants were collected 28 days post-inoculation (dpi), the nodules were counted, and the roots and shoots were separated, dried, and weighed.

### 4.7. Collection and Characterization of Soil

Soils samples were collected from the Odiel River estuary (Huelva, Spain) at two localizations, high marsh (37°15′ N, 6°58′ W; SW Spain) (OHM) and low marsh (37°13′7.00″ N, 6°57′35.92″ W) (OLM). Three samples per soil were picked between 10–20 cm of depth. Samples were chemically analyzed as described in [60]. Basically, conductivity was measured using a Crison-522 (Spain) conductivity meter and the redox potential and pH with a Crison pH/mVp-506 (Spain) portable meter. Soil texture (sand, silt, and clay percentage) was determined using the Bouyoucos hydrometer method [61]. The concentration of micro elements was determined by inductively coupled plasma–optical emission spectroscopy (ICP–OES) (ARLFisons3410, Thermo Scientific, Walthman, MA, USA).

### 4.8. Pot Inoculation in Greenhouse Conditions

For plant experiments in greenhouse conditions, plastic pots (11 cm squared pots with 12 cm heights) were filled with 1 kg of sterilized soil from the Odiel River estuary (OHM). Soil samples were sterilized in an autoclave at 121 °C for 30 min; the samples were then remixed and the sterilization repeated twice. Pre-germinated seeds were placed in pots, two seedlings per pot and eight pots per condition. For OHM soil, the inoculation conditions assayed were control without inoculation (C–), *E. medicae* MA11 (MA11), *V. paradoxus* (Vp), *V. gossypii* (Vg), co-inoculated with *E. medicae* MA11 and *V. paradoxus* (MA11 + Vp) or *V. gossypii* (MA11 + Vg), or inoculated with the three strains (CSV). Assays were also carried out in pots filled with three parts of sterile OHM soil and one part of sterile OLM. In that case, only the control plants without inoculation, inoculated with MA11, and inoculated with the three strains (CSV) were included. Plants were grown under greenhouse conditions and irrigated or inoculated with sterile water or the corresponding bacteria every week. The greenhouse had controlled light and temperature conditions; natural light was supplemented with fluorescent/incandescent lamps to get a photoperiod of 16 h light:8 h dark, and the temperature was adjusted to 25 °C during the day and 15 °C during the night. Plants were removed after 60 days and their size, dry weight, number and size of leaves, and number of nodules were recorded. Nitrogen content was determined using an InfrAlyzer 300 (Technicon, Tarrytown, NY, USA) as described in [62]. The amounts of microelements in plant tissues were measured by inductively coupled plasma–optical emission spectroscopy (ICP–OES) (ARLFisons3410, Thermo Scientific, Walthman, MA, USA).

### 4.9. Chlorophyll Content

For total chlorophyll content determination, 50 mg of random leaves were mashed using a mortar containing a 100% acetone:0.9% saline solution (4:1; *v*/*v*) [63]. Absorbance at 652 nm was measured using a spectrophotometer (Lambda 25; PerkinElmer, Walthmam, MA, USA) in duplicate samples and the total content of chlorophyll was determined by the formula described in [64].

### 4.10. Antioxidant Enzymes Determination

Catalase (CAT), ascorbate peroxidase (APX), superoxide dismutase (SOD), and guaiacol peroxidase (GPX) activities were measured in *M. sativa* leaves and roots as described in [65]. Random leaves and roots from different plants undergoing the same treatment were collected in triplicate and stored in liquid nitrogen. In total, 500 mg of vegetal material were homogenized in an extraction buffer (50 mM sodium phosphate buffer; pH 7.6) and centrifuged at 4 °C and 9000 rpm for 20 min. Total protein concentration in the vegetal extract was measured following the Bradford method [53]. Catalase activity was determined by measuring the H_2_O_2_ disappearance at 240 nm [65]. For ascorbate peroxidase, oxidation of L-ascorbate was monitored at 290 nm and superoxide dismutase was assayed, recording pyrogallol autoxidation at 325 nm. Finally, oxidation of guaiacol was measured at 470 nm to determine guaiacol peroxidase activity. Control assays were carried out in the absence of enzymatic extract samples to determine substrate autoxidation. The enzymatic activities were expressed as units per μg of protein.

### 4.11. Statistical Analyses

Statistica software version 6.0 (Statsoft Inc., Tulsa, OK, USA) was used for the statistical analyses. Results were normalized with the Kolmogorov–Smirnov test. Results from different conditions were compared using one-way ANOVA and statistic differences established using the Fisher test.

## 5. Conclusions

In conclusion, the combination of *V. gossypii* and *V. paradoxus* with a nodule-inducing rhizobia improved alfalfa growth, nodulation, and root metal accumulation in different experimental conditions. These results indicate that the *Variovorax* strains could be used as biofertilizers in nutrient-poor, degraded, and/or metal-contaminated soils to enhance legume adaptation and phytostabilization potential in soil-recovery projects.

## Figures and Tables

**Figure 1 plants-11-01091-f001:**
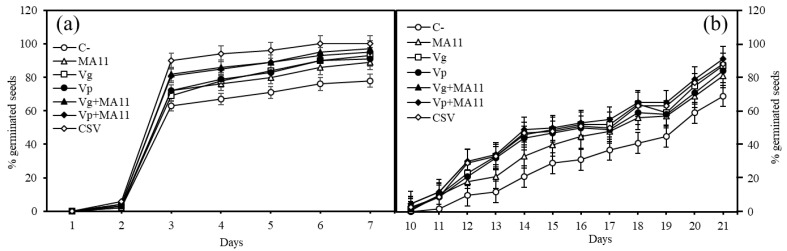
Percentage of germinated seeds. (**a**) Germination without metals and (**b**) germination with metals. Values are means ± S.D. (*n* = 50). C–: non inoculation; MA11: inoculation with *Ensifer medicae* MA11; Vg: inoculation with *Variovorax gossypii* JM-310^T^; Vp: inoculation with *V. paradoxus* S110^T^; Vg + MA11: co-inoculation with *V. gossypii* JM-310^T^ and *E. medicae* MA11; Vp + MA11: co-inoculation with *V. paradoxus* S110^T^ and *E. medicae* MA11; CSV: inoculation with *V. gossypii* JM-310^T^, *V. paradoxus* S110^T^ and *E. medicae* MA11.

**Figure 2 plants-11-01091-f002:**
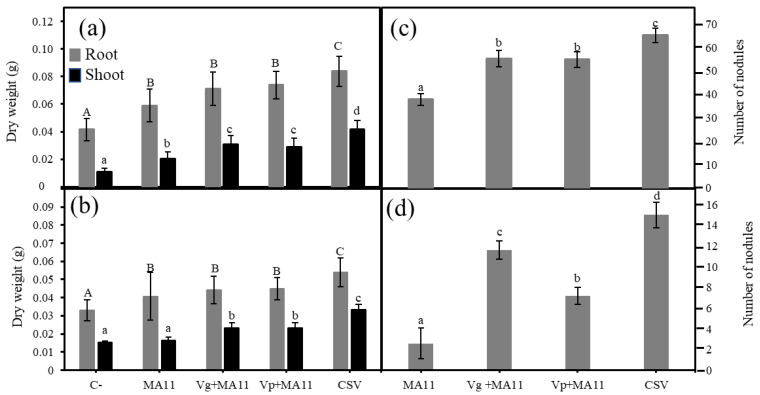
In vitro effects of *M. sativa* inoculation with *Variovorax*. (**a**) Dry weight of shoots and roots of *M. sativa* plants after 30 days in BNM plates; (**b**) dry weight of shoots and roots of *M. sativa* plants after 30 days in BNM plates supplemented with 30 μM As; (**c**) number of nodules in *M. sativa* plants after 30 days in BNM plates, (**d**) number of nodules on *M. sativa* plants after 30 days in BNM plates supplemented with 30 μM As. Values are means ± S.D. (*n* = 5). Different letters indicate means that are significantly different from each other (one-way ANOVA; LSD test, *p* < 0.0001). C–: non inoculation; MA11: inoculation with *Ensifer medicae* MA11; Vg: inoculation with *Variovorax gossypii* JM-310^T^; Vp: inoculation with *V. paradoxus* S110^T^; Vg + MA11: co-inoculation with *V. gossypii* JM-310^T^ and *E. medicae* MA11; Vp + MA11: co-inoculation with *V. paradoxus* S110^T^ and *E. medicae* MA11; CSV: inoculation with *V. gossypii* JM-310^T^, *V. paradoxus* S110^T^, and *E. medicae* MA11.

**Figure 3 plants-11-01091-f003:**
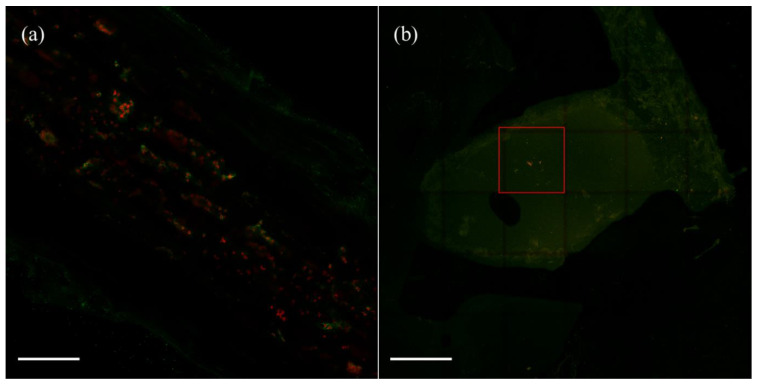
Bacterial colonization. Images of colonized *M. sativa* roots (**a**) and nodules (**b**) after 28 days of growth and inoculation with *V. paradoxus* S110^T^ marked with mCherry. Scale bars represent 600 μm in (**a**) and 1000 μm in (**b**). Red square in (**b**) marks a group of *V. paradoxus* S110^T^ marked with mCherry in the nitrogen fixation zone.

**Figure 4 plants-11-01091-f004:**
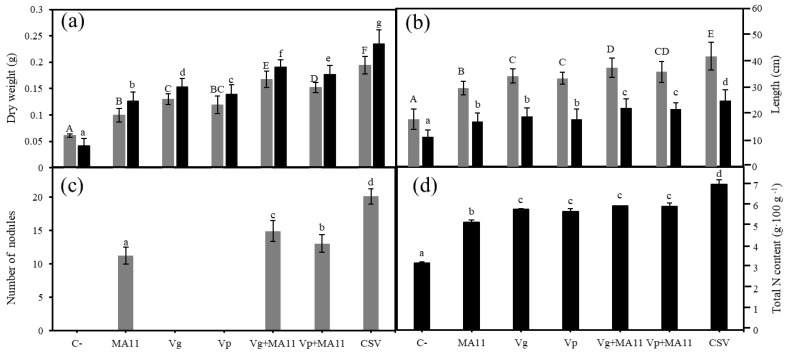
Effects of *M. sativa* inoculation with *Variovorax* in nutrient-poor soils. (**a**) Dry weight, (**b**) length, (**c**) number of nodules, and (**d**) nitrogen content in shoots of *M. sativa* plants after 60 days in pots with a nutrient-poor soil as a substrate under greenhouse conditions. Values are means ± S.D. (*n* = 16). Different letters indicate means that are significantly different from each other (one-way ANOVA; LSD test, *p* < 0.0001). C–: non inoculation; MA11: inoculation with *Ensifer medicae* MA11; Vg: inoculation with *Variovorax gossypii* JM-310^T^; Vp: inoculation with *V. paradoxus* S110^T^; Vg + MA11: co-inoculation with *V. gossypii* JM-310^T^ and *E. medicae* MA11; Vp + MA11: co-inoculation with *V. paradoxus* S110^T^ and *E. medicae* MA11; CSV: inoculation with *V. gossypii* JM-310^T^, *V. paradoxus* S110^T^, and *E. medicae* MA11.

**Figure 5 plants-11-01091-f005:**
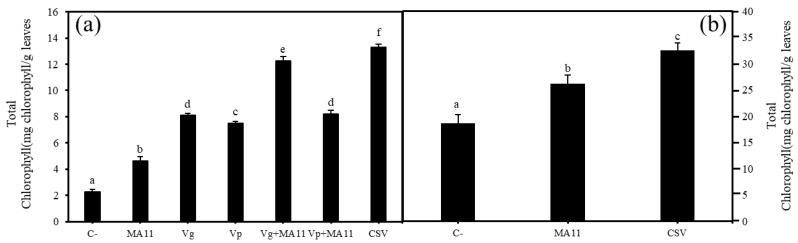
Total chlorophyll content. (**a**) The total chlorophyll content in leaves of *M. sativa* after 60 days under greenhouse conditions with a nutrient-poor soil as a substrate; (**b**) total chlorophyll content in leaves of *M. sativa* after 60 days under greenhouse conditions with soils polluted with metals. Values are means ± S.D. (*n* = 16). Different letters indicate means that are significantly different from each other (one-way ANOVA; LSD test, *p* < 0.0001). C-: non inoculation; MA11: inoculation with *E. medicae* MA11; Vg: inoculation with *V. gossypii* JM-310^T^; Vp: inoculation with *V. paradoxus* S110^T^; Vg + MA11: co-inoculation with *V. gossypii* JM-310^T^ and *E. medicae* MA11; Vp + MA11: co-inoculation with *V. paradoxus* S110^T^ and *E. medicae* MA11; CSV: inoculation with *V. gossypii* JM-310^T^, *V. paradoxus* S110^T^, and *E. medicae* MA11.

**Figure 6 plants-11-01091-f006:**
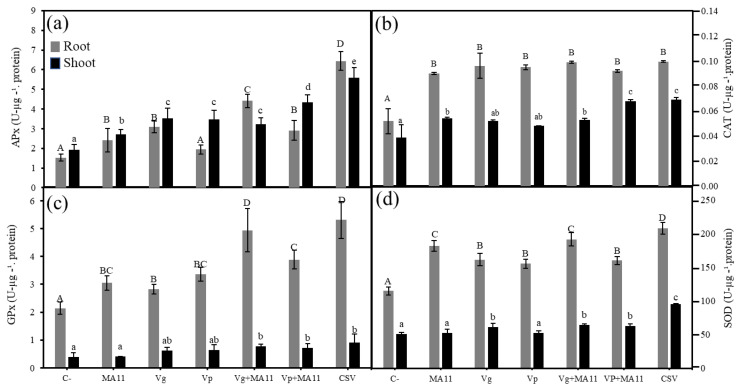
Antioxidant enzymes levels. (**a**) Ascorbate peroxidase, (**b**) catalase, (**c**) guaiacol peroxidase, and (**d**) superoxide dismutase activities in the leaves and roots of *M. sativa* after 60 days under greenhouse conditions with a nutrient-poor soil as a substrate. Values are means ± S.D. (*n* = 16). Different letters indicate means that are significantly different from each other (one-way ANOVA; LSD test *p* < 0.0001). C–: non inoculation; MA11: inoculation with *E. medicae* MA11; Vg: inoculation with *V. gossypii* JM-310^T^; Vp: inoculation with *V. paradoxus* S110^T^; Vg + MA11: co-inoculation with *V. gossypii* JM-310^T^ and *E. medicae* MA11; Vp + MA11: co-inoculation with *V. paradoxus* S110^T^ and *E. medicae* MA11; CSV: inoculation with *V. gossypii* JM-310^T^, *V. paradoxus* S110^T^, and *E. medicae* MA11.

**Figure 7 plants-11-01091-f007:**
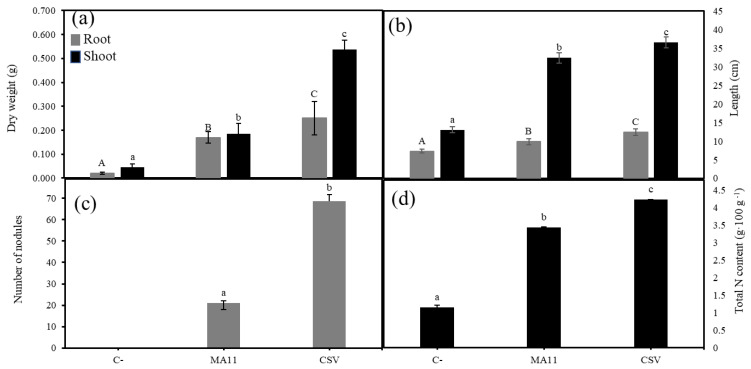
Effects of *M. sativa* inoculation with *Variovorax* in metal-contaminated soils. (**a**) Dry weight, (**b**) length, (**c**) number of nodules, and (**d**) nitrogen content in shoots of *M. sativa* plants after 60 days in pots with soils polluted with metals as substrates under greenhouse conditions. Values are means ± S.D. (*n* = 16). Different letters indicate means that are significantly different from each other (one-way ANOVA; LSD test, *p* < 0.0001). C–: non inoculation; MA11: inoculation with *E. medicae* MA11; CSV: inoculation with *V. gossypii* JM-310^T^, *V. paradoxus* S110^T^, and *E. medicae* MA11.

**Figure 8 plants-11-01091-f008:**
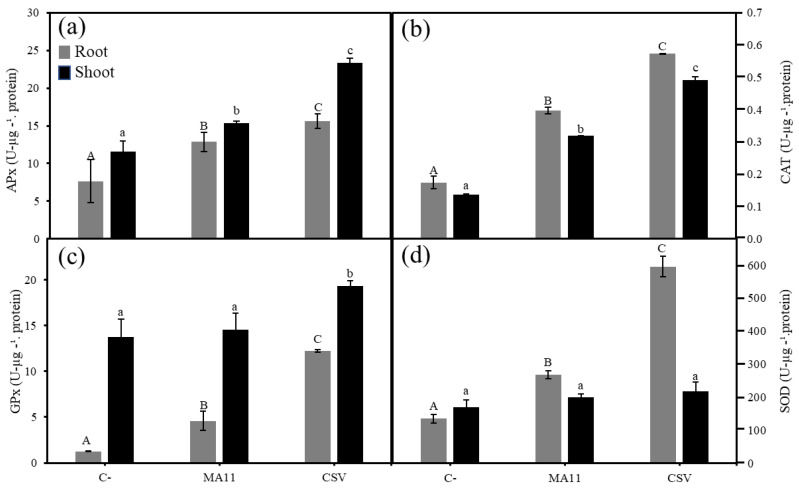
Antioxidant enzymes levels. (**a**) Ascorbate peroxidase, (**b**) catalase, (**c**) guaiacol peroxidase, and (**d**) superoxide dismutase activities in the leaves and roots of *M. sativa* after 60 days under greenhouse conditions with soil polluted with metals as substrate. Values are means ± S.D. (*n* = 16). Different letters indicate means that are significantly different from each other (one-way ANOVA; LSD test, *p* < 0.0001). C–: non inoculation; MA11: inoculation with *E. medicae* MA11; CSV: inoculation with *V. gossypii* JM-310^T^, *V*.

**Figure 9 plants-11-01091-f009:**
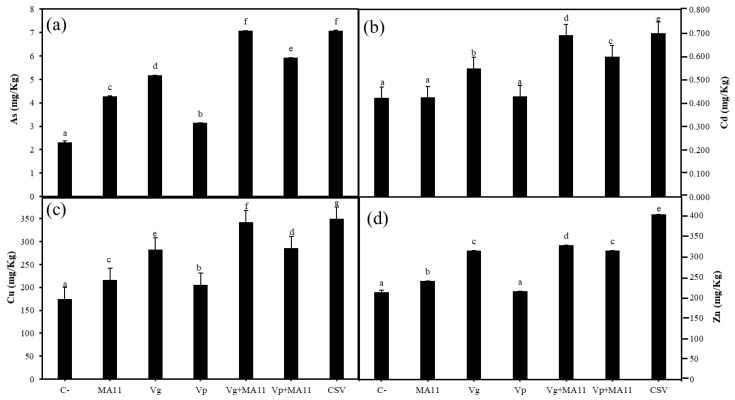
Metal/loid accumulation. (**a**) Arsenic, (**b**) cadmium, (**c**) copper, and (**d**) zinc accumulation in roots of *M. sativa* after 60 days in pots with nutrient-poor soils as a substrate under greenhouse conditions. Values are means ± S.D. (*n* = 16). Different letters indicate means that are significantly different from each other (one-way ANOVA; LSD test, *p* < 0.0001). C–: non inoculation; MA11: inoculation with *E. medicae* MA11; Vg: inoculation with *V. gossypii* JM-310^T^; Vp: inoculation with *V. paradoxus* S110^T^; Vg + MA11: co-inoculation with *V. gossypii* JM-310^T^ and *E. medicae* MA11; Vp + MA11: co-inoculation with *V. paradoxus* S110^T^ and *E. medicae* MA11; CSV: inoculation with *V. gossypii* JM-310^T^, *V. paradoxus* S110^T^, and *E. medicae* MA11.

**Table 1 plants-11-01091-t001:** Maximum tolerable concentration of metal/loid.

Strain	As (mM)	Cd (mM)	Cu (mM)	Zn (mM)
MA11	10	1.5	3	8
Vg	9	5.9	10.6	10.7
Vp	8.9	4	10.3	12

MA11: Ensifer medicae MA11; Vg: Variovorax gossypii JM-310^T^; Vp: Variovorax paradoxus S110^T^.

**Table 2 plants-11-01091-t002:** PGP properties and enzymatic activities shown by tested strains.

PGP Properties	Without Metal			With Metal		
	MA11	Vg	Vp	MA11	Vg	Vp
Phosphate solubilizing	1.70	-	-	1.10	1.20	1.00
Siderophores production	1.41	5.00	4.00	n. d	n. d	n. d
IAA production	1.76	4.26	2.11	1.07	2.28	1.51
Biofilm formation	0.14	-	0.56	0.27	0.10	0.18
ACC deaminase activity	1.03	2.66	2.11	9.08	10.03	8.55
**Enzymatic activities**						
DNAse	-	+	-	-	-	-
Amylase	+	+	+	+	-	-
Cellulase	-	-	-	-	-	-
Lipase	-	+	-	-	-	-
Pectinas	-	+	-	-	-	-
Protease	-	+	-	-	+	-
Chitinase	-	-	-	-	-	-

MA11, *E. medicae* MA11; Vg, *V. gossypii* JM-310^T^; Vp, *V. paradoxus* S110^T^; +, presence of the activity; -, absence of the property/activity; n. d., non-determinated. Values of phosphate solubilization and siderophores production express the diameter of the halo in cm. Values of IAA production are expressed in mg·L^−1^. Values of ACC deaminase activity are expressed in μmoles α-ketobutyrate·mg protein^−1^·h^−1^.

**Table 3 plants-11-01091-t003:** Physicochemical properties and micronutrient concentrations of soil from the Rio Odiel estuary.

**Physicochemical Properties**								
**Location**	**Texture (%) ***		**Organic Material (%)**		**Conductivity (μS·cm^−1^)**		**pH**	
High marsh	70/15/15		0.9 ± 0.05		12.8 ± 0.6		6.9 ± 0.1	
Low marsh	28.51/44.99/26.49		3.66 ± 0.1		15.5 ± 0.2		6.8 ± 0.05	
**Metal/Loids Concentration (mg·kg^−1^)**								
**Location**	**As**	**Cd**	**Cu**	**Zn**	**Mg**	**Na**	**Fe**	** *p* **
High marsh	26.7 ± 3.6	0.36 ± 0.1	311.1 ± 14.3	375.0 ± 9.2	0.777 ± 0.02	0.359 ± 0.006	9125.78 ± 12.4	0.054 ± 0.012
Low marsh	565.9 ± 10.9	2.3 ± 0.0	1200.3 ± 36.7	2425.7 ± 4.0	1.047± 0.064	1.707 ± 0.04	84,988.68 ± 6.1	0.412 ± 0.007

Values are mean ± S.D. (*n* = 3). * Texture (sand/slit/clay percentage).

**Table 4 plants-11-01091-t004:** Metal/loids accumulation in M. sativa tissues after 60 days under greenhouse conditions with soils polluted with metals as a substrate.

	As (mg·kg^−1^)	Cd (mg·kg^−1^)	Cu (mg·kg^−1^)	Zn (mg·kg^−1^)
**Shoot**				
**C–**	7.23 ± 0.00 ^a^	0.12 ± 0.00 ^a^	36.08 ± 0.02 ^a^	57.61 ± 0.07 ^a^
**MA11**	8.97 ± 0.02 ^b^	0.27 ± 0.00 ^b^	48.51 ± 0.00 ^b^	90.92 ± 0.06 ^b^
**CSV**	6.98 ± 0.07 ^c^	0.18 ± 0.01 ^c^	50.83 ± 0.02 ^c^	68.39 ± 0.72 ^c^
**Root**				
**C–**	17.45 ± 0.41 ^a^	0.30 ± 0.01 ^a^	78.95 ± 0.07 ^a^	214.82 ± 0.49 ^a^
**MA11**	51.82 ± 1.31 ^b^	1.23 ± 0.07 ^b^	218.30 ± 0.51 ^b^	401.45 ± 0.07 ^b^
**CSV**	39.43 ± 0.70 ^c^	1.51 ± 0.01 ^c^	326.62 ± 0.72 ^c^	430.07 ± 0.01 ^c^

Values are means ±S.D. (*n* = 16). Different letters indicate means that are significantly different from each other (one-way ANOVA; LSD test *p* < 0.0001). C–: non inoculation; MA11: inoculation with *E. medicae* MA11; CSV: inoculation with *V. gossypii* JM-310^T^, *V. paradoxus* S110^T^, and *E. medicae* MA11.

## Data Availability

Not applicable.

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
