# Peer review of "Improved *Medicago sativa* Nodulation under Stress Assisted by *Variovorax* sp. Endophytes"

_plants, 2022, doi:10.3390/plants11081091_

Round 1

Reviewer 1 Report

The main things in this article you should check statistical analysis very carefully before you resubmit it again 

Line 30-35 needs reference

Fig 2 d please check the statistical analysis carefully something wrong how Vp+AM11 take c and Vg +MA11 tack b

The same with Fig 2 B just check MA11 and Vg+M11 is this right?

Fig 4 c Vg+ M11 should be c not b and Vp+M11 should be b right?

Fig 5 a please check the statistical analysis

Fig 6 also need to check

Fig 9 how Vp ( d) and Vg ( c  )

In Discussion part you did not discuss the antioxidant enzyme

Part 4.5 need references

Also part 4.6 need references

Some part did not mentioned how many replicate you used

Greenhouse experiment you did not mentioned about the conditions, the size of pots and soil volume did not mentioned

Author Response

Reviewer: The main things in this article you should check statistical analysis very carefully before you resubmit it again 

Answer: We have carefully checked statistical analysis and introduced several changes according to reviewer indications.

R: Line 30-35 needs reference

A: The following reference has been included: [1] FAO, ITPS. Status of the World’s Soil Resources (SWSR) - Main Report. Roma: FAO and Intergovernmental Technical Panel on Soils; 2015.

R: Fig 2 d please check the statistical analysis carefully something wrong how Vp+AM11 take c and Vg +MA11 tack b

The same with Fig 2 B just check MA11 and Vg+M11 is this right?

Fig 4 c Vg+ M11 should be c not b and Vp+M11 should be b right?

Fig 5 a please check the statistical analysis

Fig 6 also need to check

Fig 9 how Vp (d) and Vg (c)

A: We have followed the criteria indicated by the reviewer, beginning with letter “a” the bar with the lowest value and then following b, c, d, etc in ascending order of value. Statistics have been checked and changed where necessary.

R: In Discussion part you did not discuss the antioxidant enzyme

A: A paragraph discussing antioxidant enzymes with four new references has been included.

R: Part 4.5 need references

A: Reference has been added (number 51)

R: Also part 4.6 need references

A: Two references have been included (numbers 15 and 59)

R: Some part did not mention how many replicate you used

A: We do not know exactly which part the reviewer mention, we have revised the new version to indicate the number of replicates when necessary.

R: Greenhouse experiment you did not mention about the conditions, the size of pots and soil volume did not mentioned

A: We have included this information, greenhouse conditions, size of pots and volume of soil have been included in the revised version.

Thanks for your review and valuable comments

Reviewer 2 Report

The topic selection is sound, research mechods are right,  results are interesting, discussions are appropriate, and conclusion is acceptable. The writing is generally good but please re-check for some minor spelling errors.

Author Response

Thanks for your revision, we have changed all the grammar and spelling errors pointed in the text.

Reviewer 3 Report

Please see the attached .pdf file.

Author Response

Since the reply is too long please see the attachment

Round 2

Reviewer 1 Report

Dear Author
I truly appreciate your valuable response for the correct and comments regarding our (reviewer's) comments. I had revised the whole manuscript as well as your comments, The manuscript was perfectly revisited, and track changes well done.

Author Response

Dear reviewer. Thanks for your time and your comments. 

Reviewer 3 Report

Please see the attached .pdf file.

Author Response

Dear reviewer, thanks again for your time and comments.

We have replaced aerial part by shoots in the text and figure captions.

In addition, que we included in the discussion the sentences "V. paradoxus S110T (genome BCUT01) and V. gossypii JM-310T (genome RXOE01) genomes have genes involved in IAA synthesis" and "Although both Variovorax strains showed ACC deaminase activity, acdS gene coding this enzyme is only found in V. paradoxus genome". In that way we highlight that we could not find acdS gene in V. gossypii genome. 

Round 3

Reviewer 3 Report

I am satisfied.